# Approaches to Enhancing Gas Sensing Properties: A Review

**DOI:** 10.3390/s19071495

**Published:** 2019-03-27

**Authors:** Zhenyu Yuan, Rui Li, Fanli Meng, Junjie Zhang, Kaiyuan Zuo, Erchou Han

**Affiliations:** College of Information Science and Engineering, Northeastern University, Shenyang 110819, China; yuanzhenyu@ise.neu.edu.cn (Z.Y.); 1800726@stu.neu.edu.cn (R.L.); 1700866@stu.neu.edu.cn (J.Z.); 1800730@stu.neu.edu.cn (K.Z.); 1700723@stu.neu.edu.cn (E.H.)

**Keywords:** gas nanosensors, nanomaterials, sensing properties, mechanism

## Abstract

A gas nanosensor is an instrument that converts the information of an unknown gas (species, concentration, etc.) into other signals (for example, an electrical signal) according to certain principles, combining detection principles, material science, and processing technology. As an effective application for detecting a large number of dangerous gases, gas nanosensors have attracted extensive interest. However, their development and application are restricted because of issues such as a low response, poor selectivity, and high operation temperature, etc. To tackle these issues, various measures have been studied and will be introduced in this review, mainly including controlling the nanostructure, doping with 2D nanomaterials, decorating with noble metal nanoparticles, and forming the heterojunction. In every section, recent advances and typical research, as well mechanisms, will also be demonstrated.

## 1. Introduction

In our life, we have contact with various harmful and flammable gases, some of which cause air pollution and affect our physical health. Based on the requirement for detecting these gases effectively, gas sensors have attracted much research interest due to their advantages, including their low cost, compact structure, long lifespan, and simple circuit [1,2,3]. Furthermore, the metal oxide semiconductor (MOS) gas sensor, one resistive type sensor, has become a hotspot in the field of gas sensors.

To estimate the performances of gas sensors, some parameters have been proposed, mainly including response, operation temperature, detection limit, response/recovery time, and selectivity [4,5]. Response cannot be measured directly, but can be calculated by two parameters—$R_{a}$ and $R_{g}$, where $R_{a}$ is the resistance while exposed to air and $R_{g}$ is the resistance while exposed to aimed gas. Because the change trend of resistance is related to the type of semiconductor and aimed gas, the method of calculation is distinct. The conductivity of the n-type semiconductor increases when in contact with reducing gas; yet the conductivity of the p-type semiconductor increases in an oxidizing atmosphere. Therefore, for the n-type semiconductor, the response is $R_{g}/R_{a}$ in oxidizing gas and $R_{a}/R_{g}$ in reducing gas; for the p-type semiconductor, the condition is the opposite (as shown in Table 1). Obviously, response illustrates the changing extent of sensitive materials and further denotes the sensitivity of sensors to a certain gas. Selectivity demonstrates if the specific recognition of one certain gas can be achieved by gas sensors. The operation temperature is the temperature at which sensors obtain an optimal response and generally, the response will increase first and decrease next with the increasing of the temperature. Apart from these parameters, other indices are also defined to indicate the gas sensing properties of sensors. The detection limit refers to the lowest concentration of aimed gas that sensors can respond to, which determines the application field of gas sensors. Response time is the period from when aimed gas is injected to when the resistance reaches a stable value in aimed gas; so, recovery time is the period from when aimed gas is removed to when the resistance reaches a stable value in air. These parameters determine the application of gas sensors.

To a great degree, these indices are related to sensitive materials that play a vital role in gas sensors and even can be seen as their core. Therefore, to tackle the disadvantages of gas sensors, such as their low response, poor selectivity, high operation temperature, and long response/recovery time, approaches to more sensitive material have been studied widely over a long period in the past. With more research about material, it has been found that nanomaterial (at least one dimension is in the nanoscale, i.e., the range of 1~100 nm) is a more suitable candidate than traditional material for gas sensing because its unique properties (such as its optical property, electrical property, thermal property, magnetic property, mechanical property, etc.) resulted from specific effects, for example, the quantum confinement effect, surface effect, macroscopic quantum tunneling effect, and small size effect [6]. For gas sensing properties, nanostructure and morphology, denoting the combination and arrangement of basic units (electrons, ions, atoms, or molecules, etc.) that make up nanomaterials, are effective. As more investigations have continued, it has been found that a regular and ordered nanostructure and morphology, decorating or doping with other materials, and forming a heterojunction could exert additional influences on materials and further improve their gas sensing properties, which will be introduced in the following sections.

## 2. Controlling the Type and Morphology of Nanostructures

### 2.1. The Type and Morphology of Nanostructures

#### 2.1.1. The Type of Nanostructures

Generally, according to the number of dimensions in the nanoscale, nanostructures can be classified as one of three kinds: zero-dimensional structure (0D structure), one-dimensional structure (1D structure), and two-dimensional structure (2D structure). Besides, composite structures constructed by one or more low dimensional structures are also known as a three-dimensional structure (3D structure). The classification of nanostructures and corresponding typical morphology are shown in Table 2.

The type of nanostructure has a significant influence on the properties of nanomaterials. In other words, the same nanomaterials with different nanostructures may exhibit distinct properties and be suitable for different applications. For example, Ma et al. [19] fabricated ZnO nanorods by the hydrothermal method without any catalyst and obtained ZnO nanoflowers composed of their as-prepared nanorods. After a series of tests, they found that nanorods showed the potential of an excellent photocatalyst for decomposing methyl orange; however, nanoflowers exhibited a preferable gas sensing performance. Recently, Wei et al. [20] synthesized ${\mathrm{WO}}_{3}$ nanorods, ${\mathrm{WO}}_{3}$ nanospheres, and ${\mathrm{WO}}_{3}$ nanoflowers by employing different reagents and hydrothermal times at the temperature of 180 °C. Additionally, the test results of $C_{2}H_{2}$ sensing properties indicated that the three examples all exhibited a good linear relationship between the response and the concentration of $C_{2}H_{2}$. Moreover, for exposure to $200{\;\mathrm{ppm}\;C}_{2}H_{2}$ at 275 °C, the response/recovery time of the nanoflowers was the shortest, and their response was about 15, which was twice that of the nanorods and 1.5 times that of the nanospheres.

#### 2.1.2. The Morphology of Nanostructures

In general, various methods for synthesizing nanomaterials can be grouped into three major categories: solid phase method, vapor phase method, and liquid phase method, according to the state of reactants, as shown in Table 3. Compared to other preparation methods, the liquid phase method has been studied widely because of its moderate reaction condition and good products with less defects and a better orientation. Therefore, in the next paragraphs, nanomaterials prepared in solvent condition will be discussed.

The morphology is related to the formation of nanocrystals. In fact, the crystal formation process can be seen as the process of the solute precipitating out of the solution, of which there are three main steps: (1) Nucleation. For a certain solvent, any substance has a corresponding solubility, and when the solution is supersaturated, the solute can precipitate, forming crystal nuclei; (2) Growth. After nuclei form, they aggregate together and the degree of supersaturation becomes lower to allow nuclei formation again, and a balance is reached. Obviously, in this period, a shorter nucleation time leads to a more uniform grain size; (3) Ripening. Ostwald ripening is the period in which larger particles continue to grow, while the smaller particles become smaller and eventually dissolve, which does not happen until reactants are depleted as the reaction goes on. The concentration of solution and hydrothermal time are also important factors [28,29]. 

In the process, nucleation and growth should be separated to manufacture quantities of nanograins with a uniform size and good morphology, which can be accomplished by using high-reactivity reagents that are difficult to prepare and store. Due to the small size of nanomaterials, there are a large number of suspended and unsaturated chemical bonds on their surface and they are in a high-energy state, thus it is difficult to control their morphology and size [30]. On the contrary, preparing a “passivation layer” on the surface of the nucleated nanograin can satisfy this requirement in a moderate condition, which can be accomplished by surfactants. First of all, surfactants can aggregate orderly to form an aggregation such as a microemulsion, lipsomes, like a “micro-reactor”, which indicates that they act as soft templates. Besides, at the early stage of the crystal formation process, surfactant molecules adsorbing on the crystal surface effectively prevent the agglomeration phenomenon and make nanomaterials more stable. Moreover, due to the different binding force between the surfactant molecules and different crystal surfaces, the former will accumulate on a specific crystal surface, which will slow down the growth rate of this crystal surface and thus play a role in guiding the direction of accumulation.

For example, Zou et al. [31] prepared three kinds of group samples by adding PVP (sample 1), PEG (sample 2), and no surfactant (sample 3), respectively, into ${\mathrm{Fe}}_{2}{\left({\mathrm{MoO}}_{4}\right)}_{3}$ to detect ethanol. By choosing varied surfactants, a wide difference in the morphology and structure was generated among these samples—sample 1 consisted of uniform spherical nanoparticles, sample 2 was assembled by templates, and sample 3 was not regular and orderly, as shown in Figure 1. It can be seen that the surfactant is vital in the formation process of nanomaterials, and its dosage and species influence the results.

Recently, Ren et al. [32] have reported an ordered mesoporous Fe doped  NiO (Figure 2) based ethanol gas sensor with a high response, good selectivity, and short response/recovery time resulting from an excellent morphology and structure. In their study, polystyrene-b-poly was used to direct the growth of NiO and finally as-prepared NiO had a special dual mesoporous, high surface, as well as an interconnected crystalline structure.

### 2.2. Some Typical Structure and Morphology

#### 2.2.1. Nanorod

The high surface-to-volume ratio and effective transformation of the chemical species on the surface [33] prove that 1D materials are suitable candidates for gas sensing materials, of which the nanorod is a typical example. For example, Dewyani and co-workers [34] used a simple co-precipitation/digestion method to synthesize ${\mathrm{Co}}_{3}O_{4}$ nanorods exhibiting excellent CO sensing properties. Especially, the response time was as short as ~3–4 s and the recovery time was ~5–6 s; the operation temperature was comparatively low at 250 °C. Their nanorods demonstrated prominent potential as CO sensors.

Generally, the growth of nanorods in the form of arrays needs a platform provided by the substrate that can be made from various materials. For instance, Narayanan et al. [35] reported an $H_{2}$ sensor prepared by ZnO nanorods grown from the ZnO seed layer on microslide glass substrates [36] via a hydrothermal method. When exposed to $100{\;\mathrm{ppm}\;H}_{2}$, the sensor had a low operation temperature of 200 °C and the response was found to be 0.7484. As shown in Figure 3, Oh et al. [37] successfully fabricated vertically aligned ZnO nanorod arrays (the average diameter and length were 50 ppm and 500 ppm, respectively) based electrochemical gas sensors. Their sensors could be obtained by introducing the 20 kHz ultrasonic waves with a density of 39.5 $W/{\mathrm{cm}}^{2}$ into the mixed solution (including 0.1 M zinc nitrate hexahydrate and 0.01 M hexamethylenetetramine), into which the substrate was immerged; the substrate was made from ${\mathrm{Al}}_{2}O_{3}$ on which the Pt electrode was deposited, immediately followed by the deposition of a Zn thin film. As a result, at a relatively low operation of 250 °C, the response was up to 824% when exposed to $100{\;\mathrm{ppb}\;\mathrm{NO}}_{2}$, the detection concentration limit was as low as 10 ppb and the response was faster than previous ${\mathrm{NO}}_{2}$ gas sensors, which was attributed to the well-aligned arrangement and high areal density in their opinion.

However, some researchers have also proposed the substrate-free method. Long et al. [38] realized the synthesis of ${\mathrm{WO}}_{3}$ nanorod arrays under a simple hydrothermal condition without any substrate. Moreover, a series of measurements demonstrated that their nanorods have a good potential in ${\mathrm{NH}}_{3}$ sensing. The response can reach 8.3 while exposed to $50{\;\mathrm{ppm}\;\mathrm{NH}}_{3}$ at 200 °C.

#### 2.2.2. Nanosheet

Due to abundant active sites and strong interconnections, nanosheets have more electron transfer channels, which can improve the reaction ratio between the materials and aimed gas [39]. However, researchers often use nanosheets to assemble other nanostructures or modify them.

Li et al. [40] synthesized hollow dodecahedrons built by ${\mathrm{Co}}_{3}O_{4}$ nanosheets via a controllable two step self-templated process. Their sensitive materials not only had a good morphology, but were also suitable for being modified easily and uniformly. Furthermore, their experiments suggested that the optimal operation temperature was 100 °C for trimethylamine; after the modification of PdO, the detection limit could reach 250 ppb and the response time was as short as 4.5 s. Zhang et al. [41] prepared ZnO microflowers assembled by nanosheets with the modification of Pd nanoparticles (as shown in Figure 4), which were applied to fabricate gas sensors exhibiting enhanced selectivity, a shorter response/recovery time, and a lower working temperature. Wang et al. [42] achieved the goal of detecting $H_{2}$ at room temperature by considerng FeOCl nanosheets decorated with Au nanoparticles as the sensitive materials.

#### 2.2.3. Micro-/Nano-Structured Hollow Spheres

Hollow micro-/nano-structured materials have been investigated widely and applied to photocatalysts [43], the treatment of organic pollutants [44], the purification of plasmid DNA [45], and supercapacitors [46], etc. due to their advantages of a higher surface-to-volume ratio and shorter distance of charge transport [47], lower density, and better permeability [48] than solid structured materials. In this part, micro-/nano-structured hollow spheres will be mainly illustrated as typical 3D curved structures. Currently, hollow spheres can be synthesized by the hard-template method, soft-template method, sacrificial-template method, and template-free method. However, the first two technologies are time-consuming, so we will only talk about the latter two.

Han et al. [49] successfully synthesized ${\mathrm{In}}_{2}O_{3}$ hollow spheres by using ${\mathrm{Cu}}_{2}O$ as sacrificial templates, exhibiting short response/recovery time toward acetone gas. In their experiments, ${\mathrm{In}}_{2}O_{3}$ was obtained by calcining $\mathrm{In}{\left(\mathrm{OH}\right)}_{3}$, of which ${\mathrm{In}}^{3+}$ ions were from ${\mathrm{InCl}}_{3}$ as the indium source and the ${\mathrm{OH}}^{-}$ ions were the products of the reaction between ${\mathrm{Cu}}_{2}O$ and $S_{2}O_{3}{}^{2-}$ ions as the coordinating etchant.

Although the template-assisted method has been widely adopted to fabricate hollow spheres, the cost of materials used as a template has urged searchers to develop the template-free method. Recently, Zhai et al. [50] reported an excellent triethylamine gas sensor, the sensitive materials of which were the ${\mathrm{WO}}_{3}$ hollow microspheres assembled by nanosheets via only a simple template-free hydrothermal process, as shown in Figure 5. They found that the sensor had a better response to triethylamine than other tested gases (the response toward 50 ppm methylbenzene, 50 ppm ethanol, 50 ppm methanol, 50 ppm acetone, 50 ppm ammonia, 50 ppm triethylamine is ~2, ~2, ~2, ~4, ~1 and ~16, respectively). Furthermore, the optimal operation temperature was 220 °C and at this temperature, the sensor had a surprisingly short response time of only 1.5 s and a recovery time of 22 s, far better than other triethylamine gas sensors. They mainly attributed the prominent sensitive performance to the many active sites provided by the numerously tiny sized pores, as well as the idea of using acidic metal oxide (${\mathrm{WO}}_{3}$) to detect basic gas (triethylamine).

#### 2.2.4. Nanoflower

Due to higher surface area and more available inter space, materials with flower-like nanostructures are attracting wide attention. Wang et al. [51] have fabricated gas sensors with 3D ${\mathrm{SnO}}_{2}$ nanoflowers assembled by 2D ${\mathrm{SnO}}_{2}$ nanosheets via a simple one-pot hydrothermal method at 90 °C. In the experiment, ${\mathrm{SnCl}}_{2}\cdot 2H_{2}O$ as tin resource was reacted with NaOH and CTAB was also added into the solution to assist the forming process of the nanostructure. As shown in Figure 6, they found that the nanostructure was related to the molar ratio ${\mathrm{OH}}^{-}$ to ${\mathrm{Sn}}^{2+}$—with the molar ration increasing from 4/1 to 10/1, the thickness of nanosheets descended from 44.7 nm to 6.2 nm; from 4/1 to 8/1, the beautiful grown nanoflowers could be seen and were varied when changing the molar ratio; when the molar ratio was higher, the petals of nanoflowers became smaller and finally, only the ovary was left. The gas sensing properties were tested in the ethanol ambient at a relatively low temperature of 240 °C. When the concentration of ethanol gas was only 10 ppm, the gas sensor exhibited a considerable response of 2.8 that increased if the concentration was higher. Besides, the gas sensors showed a fast response to ethanol gas, including a response time within 25 s and a recovery time within 60 s. Furthermore, other VOCs were also tested and the results indicated that the gas sensor could satisfy the practical requirements.

Recently, Song et al. [52] reported ZnO nanoflowers (average diameter was 0.9~1 μm) without other doping or decoration (Figure 7); however their nanomaterials exhibited a good selectivity to ${\mathrm{NO}}_{2}$ and a room temperature operation. Their detection limit was low, at 0.5 ppm, and a high response could be made to 1 ppm under room temperature. Besides, the stability was also fairly good. Obviously, the excellent performance is related to the morphology, because uniform flower petals without aggregation can provide more sites for detecting ${\mathrm{NO}}_{2}$ molecules. Zhu et al. [53] synthesized ${\mathrm{SnO}}_{2}$ nanoflowers assembled by nanosheets (Figure 8). Toward $H_{2}$ at 350 °C, the response was 22 and the response/recovery time was 10/less 10 s.

#### 2.2.5. Core-Shell Structure

Since the concept of a core-shell structure was proposed, nanomaterials with this structure have been widely applied to supercapacitor electrodes [54,55], photodetection [56], ions batteries [57,58], catalysts [59,60,61], and sensing [62]. Various nanostructured and various dimensional core-shell materials have been developed and exhibited satisfying performances in gas sensing, for example, nanoparticles [63], nanorods [64,65], nanofibers [66], nanobelts [67], nanowires [68], nanoneedles [69], micro-/nano-spheres [70,71], and nanotubes [72]. Different core/shell material systems could be formed by combining different materials, such as metal oxide/metal oxide [65], noble metal/metal oxide [73], metal oxide/noble metal [63], noble metal/noble metal [74], metal oxide/organics [75,76], and inorganics/metal oxide [77], etc.

It has been demonstrated that the application of a core-shell can further improve gas sensing compared to a conventional nanostructure. For example, Park et al. [78] synthesized ${\mathrm{Ga}}_{2}O_{3}/{\mathrm{WO}}_{3}$ core-shell nanostructures composed of nanobelts, nanoparticles, and nanowires through thermal evaporation ${\mathrm{Ga}}_{2}S_{3}$ powders followed by thermal evaporation ${\mathrm{WO}}_{3}$ powders and tested the gas sensing properties of their sensors. Compared to pristine ${\mathrm{Ga}}_{2}O_{3}$ nanostructured gas sensors, when ethanol was the aimed gas, their sensors showed a stronger response, which can be illustrated by the surface depletion layer method and the potential barrier-controlled carrier-transport method; at the same time, the total sensing time (that refers to response time adding up to recovery time) was ~50 s faster than that of pristine sensors and the operation temperature had a reduction of 100 °C; in addition, their sensors had an enhanced selectivity toward ethanol gas. Runa et al. [79] prepared well-performing ${\mathrm{NO}}_{2}$ gas sensors, of which the sensitive materials were flower-like core-shell nanostructured $\mathrm{ZnO}/{\mathrm{ZnFe}}_{2}O_{4}$ via a two step hydrothermal method under mild reaction conditions. Through the gas sensing test, they found that their sensors had better performances than pure ZnO sensors toward ${\mathrm{NO}}_{2}$ gas and benefitted from the flower-like core-shell structure and the p-n heterojunction formed at the interface between ZnO and ${\mathrm{ZnFe}}_{2}O_{4}$.

For this structure, the thickness of the shell exerts an apparent influence on the gas sensing properties. Uddin et al. [80] synthesized Pd−core/Pt−shell nanotubes structured materials by a facile two step hydrothermal process and further investigated the influences of Pt film thickness on fast response $H_{2}$ sensing properties. In their experiments, the shell thickness was controlled by various growth periods and the test results suggested that the optimal Pt coverage thickness is ultimately ~ 5−10 nm. Under optimal conditions, the working temperature was as low as 150 °C.

## 3. Doping with Two-Dimensional Nanomaterials

In 2004, Geim’s research group prepared graphene that could exist stably, which broke the previous consensus that two-dimensional materials could not exist independently. With the in-depth study of the energy band structure and various properties of graphene, two-dimensional materials represented by graphene have gradually became a research hotspot. In this section, graphene, ${\mathrm{MoS}}_{2}$, and black phosphorus will be introduced as examples.

### 3.1. Graphene

Graphene refers to a single layer of graphite atoms. Its theoretical thickness is only 0.35 nm, and its light transmittance can reach 97.7%. It is the thinest material known at present, but it is also the material with the greatest strength known at present. Its plane is composed of six carbon atoms of a cellular structure, of which each carbon atom is linked together to three other adjacent carbon atoms by σ bonds and the rest of the electrons not bonded form π bonds, perpendicular to the graphene surface in order that electrons can move freely, so it has excellent electrical conductivity and its electron mobility is as high as $250,000{\mathrm{cm}}^{2}/\left(V\cdot s\right)$ [81,82,83]. In addition, graphene with a two-dimensional structure can be curled into fullerenes with a zero-dimensional structure or carbon nanotubes with a one-dimensional structure, or it can be stacked into graphite with a three-dimensional structure to form a complete carbon family (Figure 9) [84]. Due to its excellent properties, graphene and its ramifications, such as reduced graphene oxide (rGO), have been used for catalysts [85], sensing [86], supercapacitors [87], lithium batteries [88], and so on. Additionally, recently, rGO has been regarded as a promising material for modifying MOS to obtain improving gas sensing properties [89,90,91].

Li and his co-workers [92] synthesized a ${\mathrm{Zn}}_{2}{\mathrm{SnO}}_{4}$(ZTO)/rGO nanocomposite showing better ethanol sensing properties than bare ${\mathrm{Zn}}_{2}{\mathrm{SnO}}_{4}$ and investigated the influence of the ratio between ZTO/rGO. Figure 10 shows the response of four kinds of materials to 100 ppm ethanol, and it can be seen that although the operation temperature did not reduce, the response had an obvious improvement, and the best ratio between ZTO/rGO was 8:1 (in the figure, nZTO/rGO means that ZTO:rGO = n:1). Besides, the mechanism was further explained as follows. First, higher surface-to-volume and more active sites for gas adsorption were obtained by avoiding the aggregation phenomenon of ZTO nanoparticles by doping with rGO (as shown in Figure 11). Comparing Figure 11 b and c, it can be clearly seen that doping with rGO prevented the aggregation of ZTO nanoparticles to a great extent. Furthermore, the interaction between ZTO and rGO generated numerous defects and vacancies, which also provided more active sites. Additionally, because rGO has a better conductivity, its introduction can improve the electric properties of ZTO and further facilitate electronic migration.

### 3.2. $MoS_{2}$

Molybdenum disulfide (${\mathrm{MoS}}_{2}$), a typical material of transition metal dichalcogenides, is a black powder with metallic luster and is the main component of molybdenite. Its chemical property is very stable, thus it is insoluble in organic solvent, water, and dilute acid, but can react with aqua regia, hot sulfuric acid, and hot nitric acid. Different from the zero band gap of graphene, ${\mathrm{MoS}}_{2}$ has a wider band gap [93,94,95]. Because of specific physical and chemical properties, ${\mathrm{MoS}}_{2}$ is applied to a number of fields, such as photoelectrical detecting [96], gas sensing [97], catalysts [98,99], energy storage [100], lubricant [101], etc.

Figure 12 shows the specific structure of ${\mathrm{MoS}}_{2}$ [93]. ${\mathrm{MoS}}_{2}$ consists of vertically stacked layers, each of which is formed by covalently bonded Mo−S atoms, and each neighboring layer is connected by relatively weak van der Waals forces. These weak van der Waals interactions allow gas molecules to infiltrate and diffuse freely between the layers. In this way, the resistance of ${\mathrm{MoS}}_{2}$ can dramatically change with the adsorption and diffusion of gas molecules within the layers. Compared to traditional semiconductor gas sensing materials, ${\mathrm{MoS}}_{2}$ has several specific advantages, including a larger surface-to-volume ratio, higher adsorption efficiency, and more crystal defects, which lead to its significant position in the field of gas sensing. For example, Luo et al. [102] successfully synthesized Pt-activated ${\mathrm{MoS}}_{2}/{\mathrm{TiO}}_{2}$ that can detect hydrogen at 100 °C. Obviously, the excellent gas sensing properties were related to the heterojunction between ${\mathrm{MoS}}_{2}$ and ${\mathrm{TiO}}_{2}$ and the decoration with Pt.

However, ${\mathrm{MoS}}_{2}$ is insensitive to gas composed of nonpolar molecules, to which applying noble metal as a catalyst is an effective approach. Baek et al. [103] have reported a simple method for fabricating $H_{2}$ sensors with monolayer ${\mathrm{MoS}}_{2}$ functionalized by Pd. Sensors were prepared by some simple steps including dripping ${\mathrm{MoS}}_{2}$ on ${\mathrm{SiO}}_{2}$ substrate, the deposition of Pd nanodots on the forming ${\mathrm{MoS}}_{2}$ nanosheet, and finally fixing electrodes on the surface. With exposure to $1{\%\;H}_{2}$, the ${\mathrm{MoS}}_{2}$ sensors had no reaction. In contrast, the performances, including good repeatability, a short response time of 13.1 min, a short recovery time of 15.3 min, and a low detection limit of 500 ppm, of Pd-functionalized ${\mathrm{MoS}}_{2}$ sensors displayed a huge improvement due to the formation of ${\mathrm{PdH}}_{x}$ on the Pd nanodots while exposed to $H_{2}$. Besides, ${\mathrm{MoS}}_{2}$ is apt strongly to oxidizing in air, limiting its application in gas sensing.

### 3.3. Black Phosphorus

Black phosphorus (BP), the most stable allotrope of phosphorus, was first fabricated by Bridgman in 1914 [104]. However, there was a long period where BP did not receive much attention until phosphorene (monolayer phosphorus) was successfully prepared for the first time by mechanical exfoliation in 2014 [105], which can be regarded as the beginning of its extensive study and application. Similar to graphite, BP also comprises separate layers bonded to each other by weak van der Waals forces. The unique anisotropy crystal structure (Figure 13) means that BP has various specific properties and advantages [106]. In the latter part, the black phosphorus what we talk about only refers to phosphorene rather than bulk phosphorus.

BP possesses a moderate and direct band gap bridging the gap between graphene and ${\mathrm{MoS}}_{2}$. Besides, its comparatively large carrier mobility ($600\sim 1000{\;\mathrm{cm}}^{2}/\left(V\cdot s\right)$) and moderate on/off ratio ($10^{3}\sim 10^{5})$ bridge the gap between graphene with an extraordinarily high carrier mobility $\left(10^{3}\sim 10^{5}{\mathrm{cm}}^{2}/\left(V\cdot s\right)\right),$ as well as quite small on/off ratio (5~44), and TMDs with a low carrier mobility ($10\sim 500{\;\mathrm{cm}}^{2}/\left(V\cdot s\right)),$ as well as inversely large on/off ratio ($10^{4}\sim 10^{8})$ [107]. The value of these electrical parameters above are relevant to layers of BP. Owing to excellent properties including the optical property, thermal property, mechanical property, biocompatibility, etc. [108], apart from the electrical property, BP has been broadly applied to FETs [109,110,111], batteries [112,113], photodetectors [114], gas sensors [115], protease detection, and inhibitor screening [116].

Two-dimensional materials are quite ideal candidates for gas sensing, thus BP is not an exception and has even better performances resulting from higher surface-to-volume because of its puckered crystal structure. For example, Han et al. [117] fabricated ${\mathrm{TiO}}_{2}$ doped with BP via the sol-gel method. Through their experiments, they found that 5 mol% was the optimal concentration and an obvious response could happen at ~116 °C, as shown in Figure 14. BP as a dopant could not only provide more active sites, but also form a heterojunction with ${\mathrm{TiO}}_{2}$, which facilitated gas sensing properties.

## 4. Decorating with Noble Metal Nanoparticles

The term noble metal often refers to eight kinds of metal elements, such as Au, Ag, Ru, Rh, Pd, Os, Ir, and Pt, most of which have gorgeous luster and a strong chemical stability. Recently, numerous researches have verified the fact that decoration with noble metal particles facilitates sensors’ sensitive performances, including a higher response, better selectivity, and shorter response/recovery time, etc. [118,119].

In this paragraph, ZnO, a typical n-type semiconductor, will be used as an example to illustrate the working mechanism of this approach. While exposed to air, zinc oxide could adsorb the oxygen molecules on its surface, and these adsorbed oxygen molecules could then trap free electrons from the conduction band of ZnO and consequently transform to O_2_^−^, O^−^, and O^2−^ by ionizing (as shown in Equations (1)–(4) [120,121,122]), which caused the formation of the depletion layer on the sensitive material’s surface, making the conductive channel narrow and the resistance increase.
O_2_(gas)→O_2_(ads)(1)
O_2_(ads)+e^—^→O_2_^—^(2)
O_2_^—^+e^—^→2O^—^(3)
O^—^+e^—^→O^2^^—^(4)
2CO+O_2_^—^→2CO_2_+e^—^(5)
CO+O^—^→CO_2_+e^—^(6)
CO+O^2^^—^→CO_2_+2e^—^(7)

When reduced gas is diffused, it will be oxidized by oxygen ions on the surface, so the captured electrons will be released into the conduction band of ZnO (as shown in Equations (5)–(7) [120,121,122]). As a result, the thickness of the electron depletion layer of ZnO becomes thinner, the concentration of free electrons in ZnO increases, and the resistance decreases [123,124,125].

Generally, two mechanisms can illustrate the role of noble metal in enhancing gas sensing properties, i.e., chemical sensitization and electronic sensitization: (1) Chemical sensitization [126,127]. Oxygen molecules are prone to adsorbing on noble metal particles, hence these excessive molecules will further “spill” onto the surface of ZnO, increasing the number of reactive oxygen molecules involved in the reaction process and at the same time, providing more active sites, as shown in Figure 15 [127]; (2) Electronic sensitization [128,129,130]. Because zinc oxide has a lower work function than noble metal, when noble metal particles are decorated on the surface, a portion of free electrons will transfer from ZnO to those particles so that Schottky junctions will be formed at the contact interface, which can further decrease the concentration of free electrons in the conduction band of ZnO and deepen the thickness of the electron depletion layer. Besides, noble metal oxides are a big acceptor of oxygen; while making contact with reducing gas, they are also reduced to noble metal, which contributes a bunch of electrons. Also, the modification of noble metals can obviously shift the Fermi level of ZnO and reduce the required energy for electron transition, as shown in Figure 16 [131].

Noticeably, different effects will be produced if different kinds of noble metal particles are employed. Esfandiar et al. [132] reported that ${\mathrm{TiO}}_{2}$ nanoparticles decorated by Pd and Pt had an improved sensitivity toward hydrogen than pure ${\mathrm{TiO}}_{2}$. In their experiment, they first synthesized ${\mathrm{TiO}}_{2}$ nanoparticles via the sol-gel method, and then used the chemical reduction method to decorate the hybrid structures with Pd and Pt, and the results of gas sensitivity are shown in Figure 17. From the figure, it can be seen that the decoration of Pd or Pt both could improve the sensitive performances, but Pd had a better function. Particularly, the decoration of Pd nanoparticles dramatically increased the sensitivity and shortened the response time determined by the $H_{2}$ adsorption rate, which indicates opinions that are referred to above.

The concentration of noble metal particles loaded on the surface of sensitive materials is also an important factor, because the sensitivity of the gas sensor is not only related to the chemical reaction between material and gas, but is also affected by gas transport and utilization. It can be easily understood that with the continuous increasing of particles, the agglomeration phenomenon will happen, and a lot of wasted active sites and the too thick layer of noble metal particles will produce a negative influence on gas transport. Arunkumar et al. [123] prepared a highly selective and sensitive CO gas sensor by decorating Au nanoparticles on a zinc oxide (ZnO) nanostructure through the solution impregnation technique. They found that the response was higher with the changing concentration of Au nanoparticles from 1 wt % to 3 wt % and would decrease if more Au nanoparticles were applied.

In addition, the synergistic effect means that a bimetallic catalyst is a preferable option. In the past years, bimetallic catalyst systems which have been frequently investigated are Au-Pt [133,134,135], Pt-Pd [136,137,138,139,140], Au-Pd [141,142], Au-Ag [143], etc. Kim et al. [144] have reported an acetone gas sensor with excellent sensitive performances. They synthesized ${\mathrm{WO}}_{3}$ nanorods by the thermal evaporation method, and these nanorods were then submerged into the mixed solution composed of acetone, ${\mathrm{HAuCl}}_{4}$, and ${\mathrm{PdCl}}_{2}$ preceding UV irradiation and annealing. At 300 °C, the sensitive properties of four samples were examined, including pure ${\mathrm{WO}}_{3}$, $\mathrm{Au}-{\mathrm{WO}}_{3}$, $\mathrm{Pd}-{\mathrm{WO}}_{3}$, and $\mathrm{AuPd}-{\mathrm{WO}}_{3}$ (as shown in Figure 18). From Figure 18, it can be seen that ${\mathrm{WO}}_{3}$ nanorods after being decorated with noble metals and the synergistic effect between two kinds of noble metals can obtain a better result. Although the response and response time were both improved, an obvious change did not occur with respect to the recovery time.

## 5. Forming the Heterojunction

The heterojunction refers to the contact transition region formed at the interface between two semiconductors with different band gap widths. As we all know, semiconductors consist of p-type semiconductors (hole concentration > free electron concentration, such as ${\mathrm{Cu}}_{2}O$, NiO, ${\mathrm{VO}}_{2}$) and n-type semiconductors (free electron concentration > hole concentration, such as ZnO,${\mathrm{Ta}}_{2}O_{5}$). Therefore, a heterojunction can be classified as a homogeneous heterojunction (n-n heterojunction and p-p heterojunction) and heteromorphic heterojunction (p-n heterojunction). Heterojunctions may occur in various structures, such as nanoparticles [145], nanosheets [146], flower-like hollow microspheres [147], core-shell structured nanofibers [148]/nanorods [149]/nanoneedles [150], etc.

Due to the different chemical parameters and physical parameters, including band structure, dielectric constant, lattice constant, and electron affinity, etc. of the two materials, the mismatch phenomenon at the interface endows the heterojunction with a lot of novel properties, which have been attracting much research on heterogeneous materials and devices. In the past, heterojunctions have made tremendous contributions to photocatalysts [151,152,153], solar cells [154], photovoltaic applications [155], photoelectrochemical applications [156,157,158], diode devices [159], etc. The unique properties of a heterojunction also make it an effective and practical candidate for gas sensors.

### 5.1. n-n Heterojunction

In 2012, Sharma and his colleagues [160] used sensors made from ${\mathrm{SnO}}_{2}$ film loaded with ${\mathrm{WO}}_{3}$ micro-discs to detect trace ${\mathrm{NO}}_{2}$ gas and obtained a good result. Compared to pure ${\mathrm{SnO}}_{2}$ or ${\mathrm{SnO}}_{2}$ and ${\mathrm{WO}}_{3}$ compounds, their sensors had a higher response of 54,000 and lower optimal temperature of 100 °C, as well as a fast response/recovery time of 67 s/17 min. They supposed that the space charge region caused by n-n heterojunctions played an important role in the enhancement of ${\mathrm{NO}}_{2}$ sensing properties.

### 5.2. p-p Heterojunction

Alali et al. [161] synthesized $\mathrm{CuO}/{\mathrm{CuCo}}_{2}O_{4}$ nanotubes via the electrospinning method and a subsequent heat treatment process. Those nanotubes containing the p-p heterojunction could detect n-propanol at room temperature and showed a response of 14 and a short response/recovery time of 6.3 s/4.1 s. At the same time, those nanotubes also had a better selectivity toward n-propanol than other gases and a good stability.

Tai et al. [162] developed an ${\mathrm{NH}}_{3}$ gas sensor working at room temperature by using the microstructure silicon arrays that polyaniline nanorods were deposited on. The unique 3D structure and the formation of p-p heterojunction brought huge advantages in ${\mathrm{NH}}_{3}$ sensing.

### 5.3. p-n Heterojunction

Gong and his colleagues [163] reported an ultrasensitive ${\mathrm{NH}}_{3}$ gas sensor from n-type semiconductor ${\mathrm{TiO}}_{2}$ microfibers attached by a p-type polymer polyaniline nanograin for the first time. According to their test results, the detection limit was as low as 50 ppt in air and under this condition, a high response could also be obtained. Noticeably, there had never been reports about tens of parts-per-trillion levels of ${\mathrm{NH}}_{3}$ gas detection before their work.

Huang et al. [164] demonstrated a special heterojunction, i.e., an n-p-n heterojunction, existing in the contact region between ZnO film and ${\mathrm{SnO}}_{2}$ nanorods. In their experiment, their sensitive materials could be obtained by the preparation of pristine ${\mathrm{SnO}}_{2}$ nanorod arrays via the plasma-enhanced CVD method at low temperature and then spin coating in the zinc acetate ethanol solution to form one ZnO layer on the surface of nanorods. Toward $100{\;\mathrm{ppm}\;H}_{2}$—one typical reduced gas, totally opposite to the n-type sensing response of pristine ${\mathrm{SnO}}_{2}$ sensors, an abnormal p-type sensing response occurred in terms of their sensors and the maximum response of 18.4 could be obtained at the optimal operation temperature of 350 °C. However, when other reduced gas, including CO, ${\mathrm{NH}}_{3}$, and ${\mathrm{CH}}_{4}$, was injected, a reverse trend was displayed for the response to $H_{2}$. Additionally, they also found that the response type was related to the concentration and the humidity. The formation of the $n-\mathrm{ZnO}/p-\mathrm{Zn}-O-\mathrm{Sn}/n-{\;\mathrm{SnO}}_{2}$ heterojunction together with the unique nanostructure contributed to the potential application of exclusive $H_{2}$ sensors.

## 6. Conclusions

In this review, four kinds of measures tackling issues restricting the application and performance of gas nanosensors are mainly introduced, including controlling the nanostructure, doping with two-dimensional nanomaterials, decorating with noble metal nanoparticles, and forming the heterojunction.

Generally, a better nanostructure and morphology is very important to gas sensing properties and can even be regarded as the base of an excellent performance, which can be accomplished by surfactant. However, additional surfactant leads to the influence of humidity on gas sensing, which is acceptable. Because the enhancing effect caused by the nanostructure is finite, to further improve the sensing properties and satisfy more requirements, additional decoration or dope are developed and combined with an ordered structure and morphology. Compared to other methods, the preparation process of 2D nanomaterials is harder, the cost is higher, the period is longer, and they are prone to be oxidized in air, which is difficult to solve, so the other two methods are preferred by researchers due to their comparatively moderate preparation condition and proven technology. Noble metal has been employed widely due to its unique performance and plays an important role in gas sensors. Besides, the high price can be compensated for by their excellent enhancing effect. Due to the distinction of the performance in different materials, the heterojunction is very wide and its preparation is comparatively simple and the cost is not too high.

Although many approaches have been put forward, there is still much room for improvement. In terms of sensing gas, in my opinion, it is most important that materials must be stable in gas because of its performance or can be improved by some processes. Only in this way can it be practical and useful. At the same time, large surface-to-volume and not too difficult preparation and so on are also significant. More materials suitable for gas sensors should be excavated and through these enhancing approaches, can contribute a lot to gas nanosensors. Additionally, a more reasonable model for the gas sensing mechanism should also be connected to develop it better.

## Figures and Tables

**Figure 1 sensors-19-01495-f001:**
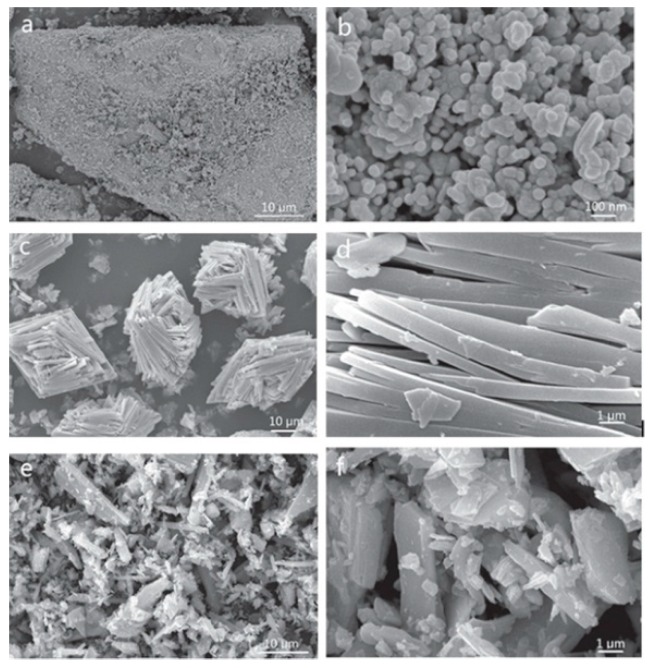
SEM images of (**a**,**b**) sample 1; (**c**,**d**) sample 2; and (**e**,**f**) sample 3 [31].

**Figure 2 sensors-19-01495-f002:**
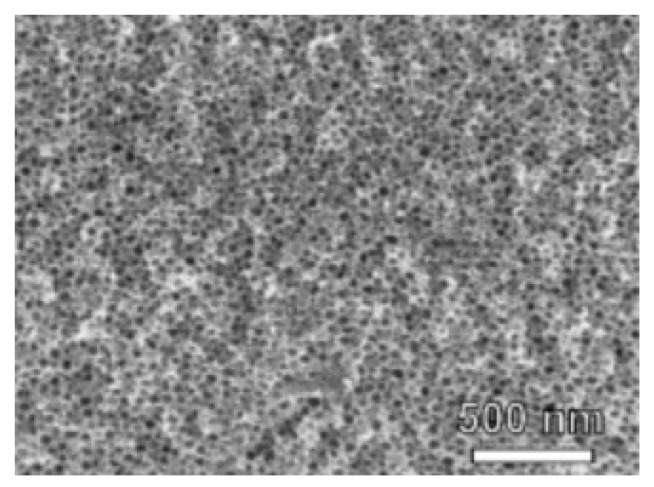
The SEM images of mNiO after calcination [32].

**Figure 3 sensors-19-01495-f003:**
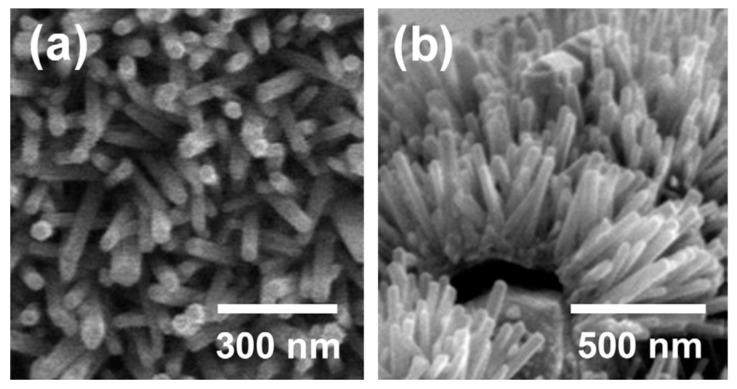
(**a**) Top view and (**b**) oblique view of SEM images of ZnO nanorod arrays [37].

**Figure 4 sensors-19-01495-f004:**
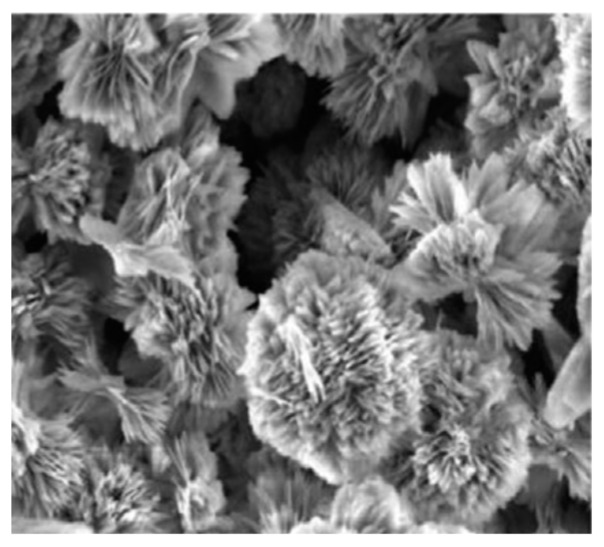
FESEM of ZnO micro flowers assembled by nanosheets [41].

**Figure 5 sensors-19-01495-f005:**
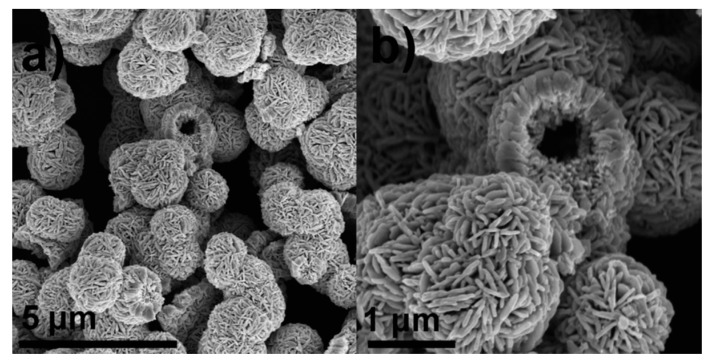
SEM image of ${\mathrm{WO}}_{3}$ hollow microspheres when the resolution is: (**a**) 5 μm and (**b**) 1 μm  [50].

**Figure 6 sensors-19-01495-f006:**
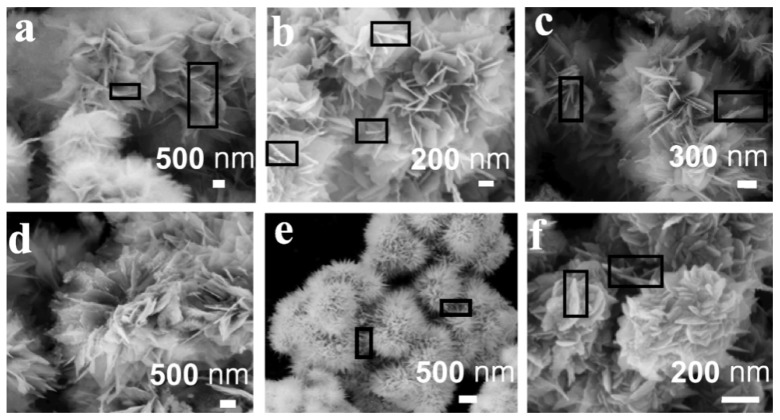
FESEM images of ${\mathrm{SnO}}_{2}$ nanoflowers when the molar ratio ${\mathrm{OH}}^{-}$ to ${\mathrm{Sn}}^{2+}$ was (**a**) 4/1, (**b**) 5/1, (**c**) 6/1, (**d**) 7/1, (**e**) 8/1, and (**f**) 10/1, respectively [51].

**Figure 7 sensors-19-01495-f007:**
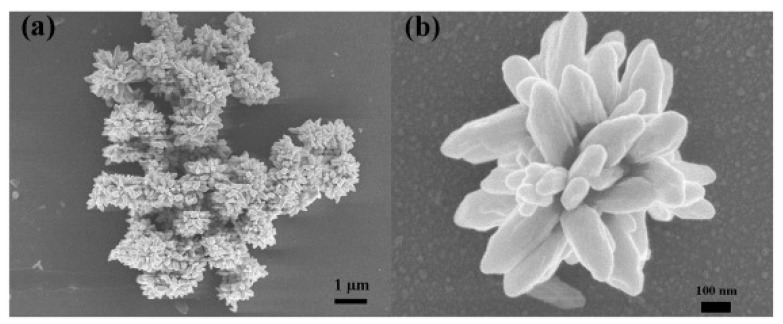
SEM images of ZnO nanoflowers with the resolution of (**a**) 1 μm and (**b**) 100 nm [52].

**Figure 8 sensors-19-01495-f008:**
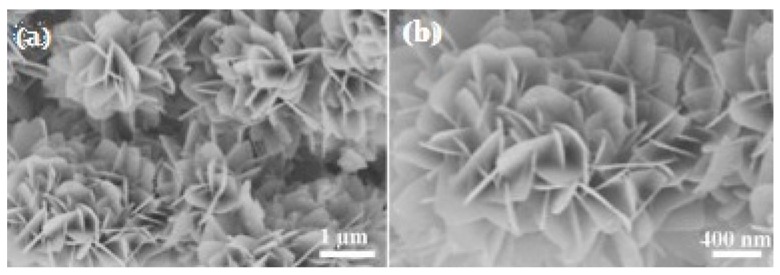
SEM images of ZnO nanoflowers with the resolution of (**a**) 1 μm and (**b**) 400 nm [53].

**Figure 9 sensors-19-01495-f009:**
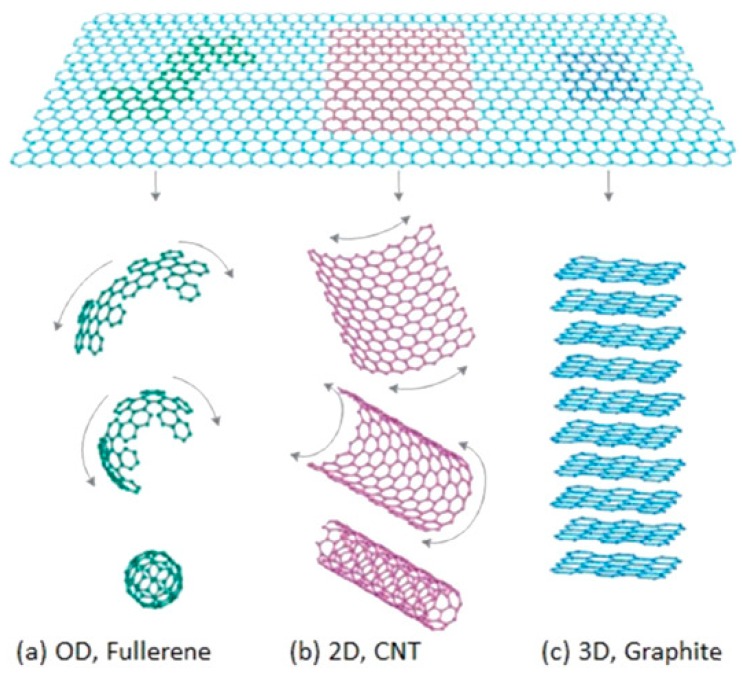
Graphene nanosheets can be transformed to (**a**) fullerene, (**b**) carbon nanotube, and (**c**) graphite [84].

**Figure 10 sensors-19-01495-f010:**
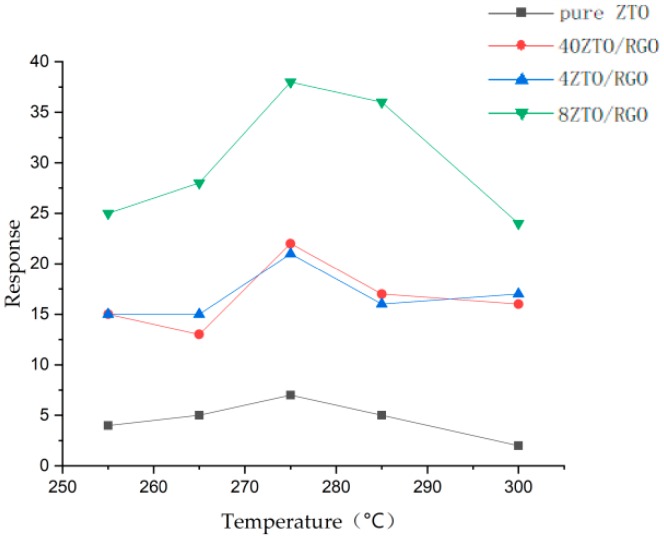
The response of four kinds of sample to to 100 ppm ethanol.

**Figure 11 sensors-19-01495-f011:**
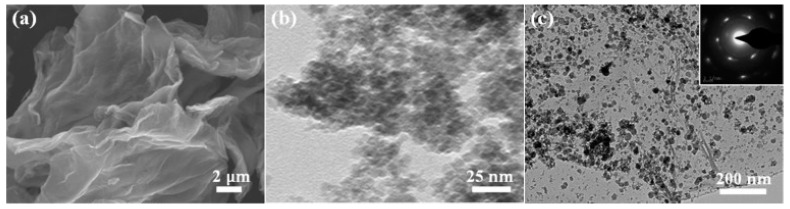
SEM images of (**a**) rGO, (**b**) ZTO nanoparticles, and (**c**) 8ZTO/rGO [92].

**Figure 12 sensors-19-01495-f012:**
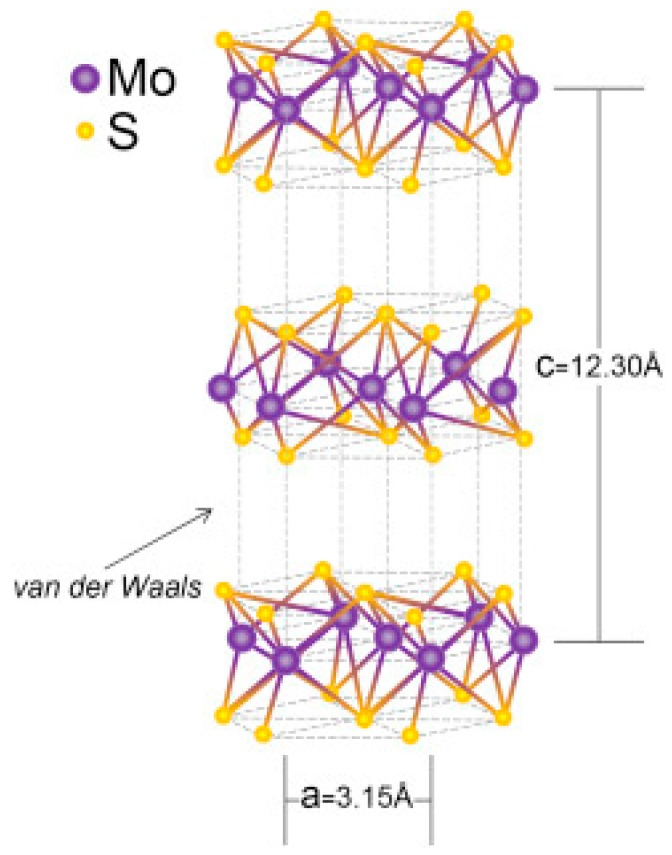
Schematic diagram for the structure of MoS_2_ [93]_._

**Figure 13 sensors-19-01495-f013:**
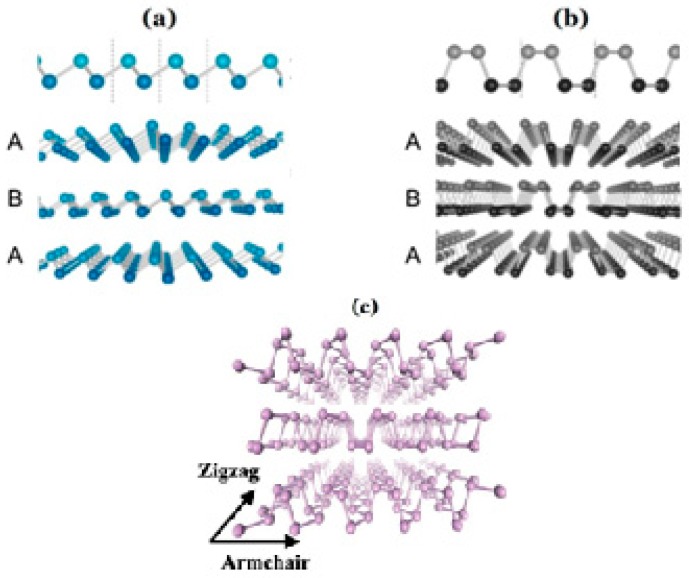
(**a**) Zigzag direction, (**b**) armchair direction, and (**c**) straight view of black phosphorus.

**Figure 14 sensors-19-01495-f014:**
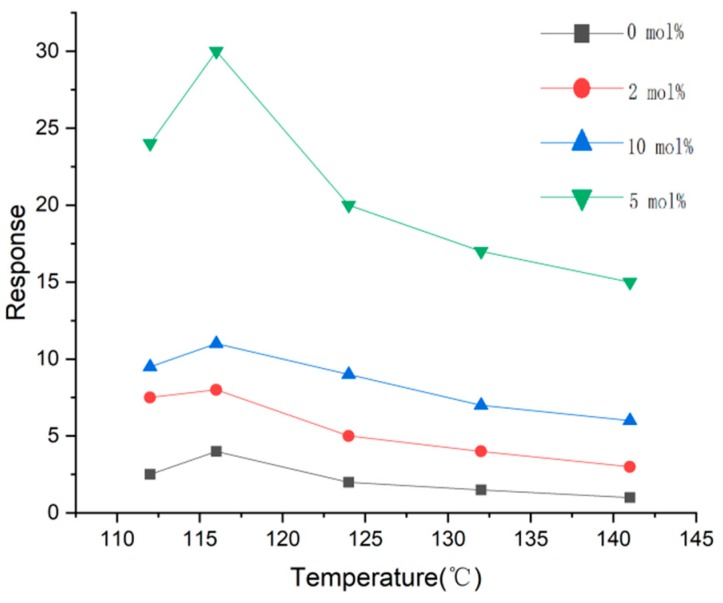
Sensitivity of ${\mathrm{TiO}}_{2}$/BP (varied concentration) to 100 ppm oxygen gas.

**Figure 15 sensors-19-01495-f015:**
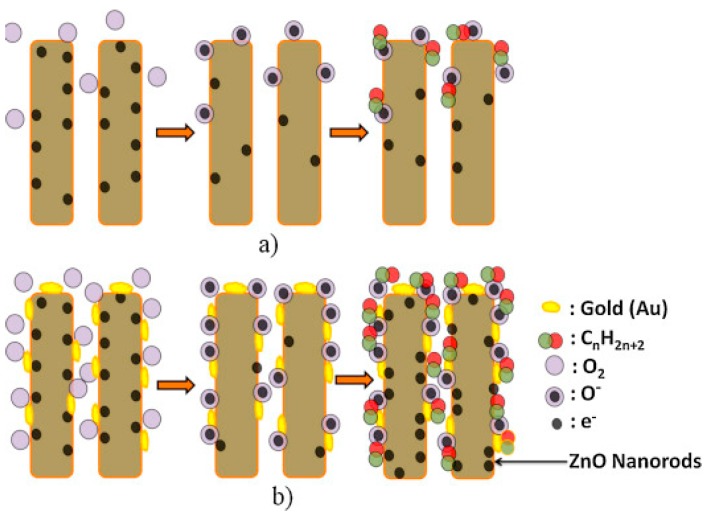
Schematic diagram of alkanes sensing of (**a**) pure ZnO nanorods and (**b**) Au functionalized nanorods [127].

**Figure 16 sensors-19-01495-f016:**
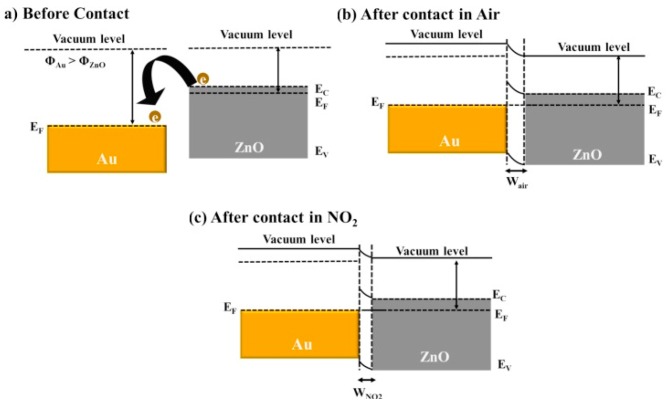
Schematic diagram of energy level of Au functionalized ZnO (**a**) before contact with air, (**b**) during contact with air, and (**c**) during contact with NO_2_ [131].

**Figure 17 sensors-19-01495-f017:**
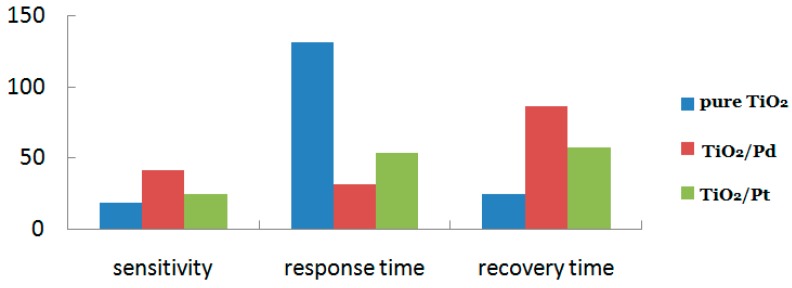
Results of different samples toward $500{\;\mathrm{ppm}\;H}_{2}$ at 180 °C in Esfandiar’s experiments.

**Figure 18 sensors-19-01495-f018:**
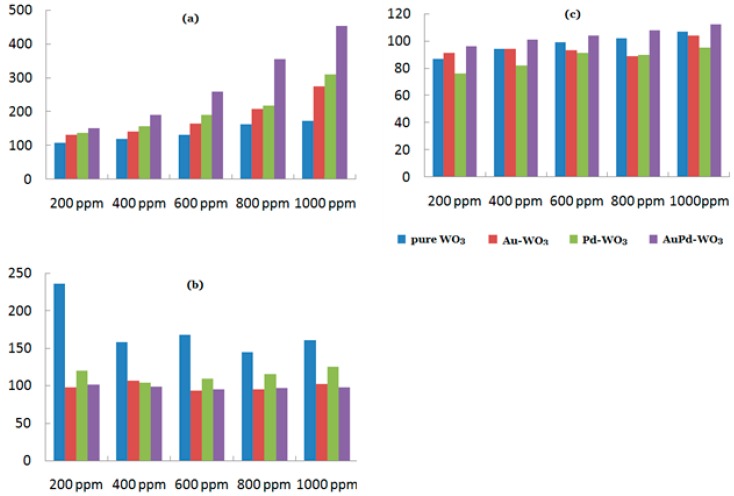
(**a**) Response, (**b**) response time, and (**c**) recovery time of different samples toward acetone at 300 °C.

**Table 1 sensors-19-01495-t001:** Response of gas sensors in different conditions.

Type of Sensitive Material	Type of Aimed Gas	Response(S)
n-type	oxidizing	$R_{g}/R_{a}$
	reducing	$R_{a}/R_{g}$
p-type	oxidizing	$R_{a}/R_{g}$
	reducing	$R_{g}/R_{a}$

**Table 2 sensors-19-01495-t002:** The classification of nanostructures and typical morphology.

	The Character of Structure	Typical Morphology
0D structure	three dimensions are in the nanoscale	Nanoparticle [7], quantum dot [8], nanocluster [9]
1D structure	two dimensions are in the nanoscale	Nanowire [10], nanofiber [11], nanorod [12], nanotube [13,14]
2D structure	one dimension is in the nanoscale	Nanosheet [15], nanobelt [16], superlattice [17]
3D structure	assembled by one kind or more low dimensional materials	Nanoflowers [18]

**Table 3 sensors-19-01495-t003:** The type and characters of preparation methods of nanomaterials.

Method	Advantages	Disadvantages	Examples
solid phase method	simple synthesis process, high yield, less pollution	uneven distribution of particle size, high agglomeration	ball milling method [21], shear milling method [22]
vapor phase method	high purity powder, small particle size, less agglomeration	high cost, high requirements for instruments	molecular beam epitaxy [23], cathode sputtering [24]
liquid phase method	simple synthesis process, controllable particle size	low distribution, low uniformity	sol-gel method [25], micro-emulsion method [26], hydrothermal method [27]

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
