# Peer review of "Approaches to Enhancing Gas Sensing Properties: A Review"

_sensors, 2019, doi:10.3390/s19071495_

Round 1

Reviewer 1 Report

You claim to provide a review on approaches to the improvement of gas sensing properties. In general, when I read your manuscript I get a feeling of reading just a bunch of facts. You describe the types of nanomaterials but don’t say how to control the growth (how to get to a desired type”. You bring examples but don’t (or insufficiently) interpret or interconnect them… I somehow just miss a structure in your manuscript… when reading the title “Approaches to the improvement of gas sensing properties: A review” I hope to find out 1) what possibilities do I have and 2) what should I do in order to get which result. You provide some details on the possibilities and give some examples. But If I wanted to know what to do to increase e.g. the response to an oxidizing gas it would be difficult to impossible to extract this information from your review… The situation becomes even more complicated because of the English language and style… It is often very difficult to understand what you are trying to say. Overall, I think you have to rewrite the manuscript. First you should think about the purpose for the reader. If you were the one who wants to know about the approaches to the improvement of gas sensing properties, what would be the most important information for you? Which details and sources would be the most important? This will give you an idea for an appropriate structure. Then you should write the manuscript and let someone with good English skills check it. After having done that your manuscript might be ready for reviewing and publishing.

Some specific comments:

Lines 9-16 (abstract): unfortunately, the abstract is written in bad non-scientific English… E.g: you don’t start a sentence with “how” in case it is not a question; you do not write “a worthy topic” in a scientific publication; “obstacles limit the application and utilization of gas sensors mainly include” is grammatically wrong – it should be “obstacles limiting”. Besides, obstacles don’t limit. You either have obstacles or limitations… Further on, what do you mean by “controlling the nanostructure” or “decoration of noble metals”? In my opinion you should rewrite the abstract from scratch.

Lines 19-48 (Introduction): sources are missing. You definitely used some literature to provide the facts given in your introduction. Also your English skills seem to be not sufficient for a scientific publication. You SHOULD let a native speaker (or at least someone who has adequate English skills) check your paper before submitting it the next time. I assume that the rest of the paper is written in the same manner, so I will not comment on English any more. Also I have a feeling that you do not 100% understand what you are writing about… maybe it is because of your way of presenting the information…

Lines 31-33: not “reduced” and “oxide” bur “reducing” and “oxidizing”. Further on, you talk about the resistance but show the sensitivity S…

Lines 39-40: where did you read this? This definition seems odd to me.

Line 49: maybe you wanted to say “controlling the type and morphology of the nanostructures”?

Table 2: source?

Line 57: the TYPE of the nanostructure

Lines 68-69: how do you calculate when saying “twice”? the response for nanorods was 20s, for nanospheres – 14 s and for nanoflowers – 12 s. After that you say the response for nanoflowers (again) was about 15. What is true?! I think you lost yourself in this data…

Lines 213-216: what do all these abbreviations stand for?

Chapter 2.2: so how does the type of the surfactant influence the morphology and grain size? Are there any dependencies? You only write there IS an influence but it is by far more interesting, WHICH surfactant influences IN WHAT WAY…

Chapter 3.1: sources for your information about graphene?

Line 240: are you SURE fullerenes are zero-dimensional? I think they are 3D…

Chapter 3: your chapter seems to deal with decorating 2D nanomaterials but you only describe 3 materials – graphene, MoS2 and BP. And only for MoS2 you talk about decorating it with Pd nanodots… either the header for this chapter is wrong or you provide wrong details…

Chapter 4: I suppose you do not mean “decoration OF noble matal nanoparticles” but “decorating WITH…”. This is an interesting and important chapter – you should rewrite it in a clear and structured way…

Equations 1 and 2: source?

Chapter 6: you call it a conclusion but provide just a summary or even a kind of “table of contents”. The CONCLUSION is missing. You don’t even try to analyse different methods, to compare the results obtained or to find out dependencies…

Author Response

Dear reviewer1:

I am very grateful to your comments for the manuscript which not only enhance our paper but also helpful for our future research. Based on your comments and requests, we have made extensive modification on the original manuscript. A revised manuscript with the correction sections red marked was attached as the supplemental material and for easy check/editing purpose. We really hope to have the opportunity to publish our article in Sensors, and we will be love to revise the article further if it is still not acceptable.

Should you have any questions, please contact us without hesitate. Now I answer the questions one-by-one in attached PDF.

Thank you very much.

Sincerely,

Zhenyu Yuan

Reviewer 2 Report

See the attached reviewing report.

Author Response

Dear reviewer 2:

I am very grateful to your comments for the manuscript which not only enhance our paper but also helpful for our future research. Based on your comments and requests, we have made extensive modification on the original manuscript. A revised manuscript with the correction sections red marked was attached as the supplemental material and for easy check/editing purpose. We really hope to have the opportunity to publish our article in Sensors, and we will be love to revise the article further if it is still not acceptable.

Should you have any questions, please contact us without hesitate. Now I answer the questions one-by-one in attached PDF.

Thank you very much.

Sincerely,

Zhenyu Yuan

Reviewer 3 Report

The authors review the litterature for gas sensing with respect to nanomaterials and controlling nanostructures and morphology to improve the gas sensing figure of merits. The review is systematic written and with an acceptable critical attitude. However, the litterature is huge, so an optional to the authors is if they want to add more relevant references it can strengthen the review.

Author Response

Dear reviewer 3:

I am very grateful to your comments for the manuscript which not only enhance our paper but also helpful for our future research. Based on your comments and requests, we have made extensive modification on the original manuscript. A revised manuscript with the correction sections red marked was attached as the supplemental material and for easy check/editing purpose. We really hope to have the opportunity to publish our article in Sensors, and we will be love to revise the article further if it is still not acceptable.

Should you have any questions, please contact us without hesitate. Now I answer the questions one-by-one.

Thank you very much.

Sincerely,

Zhenyu Yuan

Response to Reviewer 3

Your ideas remind me of what an article was needed when I began to touch gas nanosensors. Putting myself in the position of the fresh, I struggle to revise this review, and some big modifications mainly include as follows:

(1) Abstract, introduction and conclusion are reconsidered and rewrote.

(2) In section 2, growth method of grain is introduced and more recent advances in nanoflowers are added.

(3) In section 3, some diagrams of materials are added and advances are reselected.

(4) In section 4, an investigation was remade and the mechanism has been introduced different from the manuscript and some mechanism diagrams are also added.

        As for your detailed advice, more references have been added into this article.

Reviewer 4 Report

This is a good and timely review of gas sensors and approaches to improve their performances. The manuscript reads well though I feel there is a lack of graphical presentation of data. Tables are fine but some of these tables could be presented in graphical display and that would give a quick over view of the progress of different sensors.

Author Response

Dear reviewer 4:

I am very grateful to your comments for the manuscript which not only enhance our paper but also helpful for our future research. Based on your comments and requests, we have made extensive modification on the original manuscript. A revised manuscript with the correction sections red marked was attached as the supplemental material and for easy check/editing purpose. We really hope to have the opportunity to publish our article in Sensors, and we will be love to revise the article further if it is still not acceptable.

Should you have any questions, please contact us without hesitate. Now I answer the questions one-by-one.

Thank you very much.

Sincerely,

Zhenyu Yuan

Response to Reviewer 4

Your ideas remind me of what an article was needed when I began to touch gas nanosensors. Putting myself in the position of the fresh, I struggle to revise this review, and some big modifications mainly include as follows:

(1) Abstract, introduction and conclusion are reconsidered and rewrote.

(2) In section 2, growth method of grain is introduced and more recent advances in nanoflowers are added.

(3) In section 3, some diagrams of materials are added and advances are reselected.

(4) In section 4, an investigation was remade and the mechanism has been introduced different from the manuscript and some mechanism diagrams are also added.

 As for your detailed advice, Some graphs are added.(Figure 17 and 18)

Round 2

Reviewer 2 Report

The manuscript can be accepted for publication.